# Efficacy of UVC Radiation in Reducing Bacterial Load on Dental Office Surfaces

**DOI:** 10.3390/dj13120596

**Published:** 2025-12-12

**Authors:** Souat Tsolak, Eugen Bud, Sorana Maria Bucur, Mariana Păcurar, Adrian Man, Daniela Manuc

**Affiliations:** 1Doctoral School, Romanian Academy, 010071 Bucharest, Romania; tsolaksouat@gmail.com; 2Department of Orthodontics, George Emil Palade University of Medicine, Pharmacy, Science, and Technology of Târgu Mureș, 38 Ghe. Marinescu Street, 540139 Târgu Mureș, Romania; eugen.bud@umfst.ro; 3Department of Dentistry, Faculty of Medicine, “Dimitrie Cantemir” University of Târgu Mureș, 540545 Târgu Mureș, Romania; bucursoranamaria@gmail.com; 4Department of Microbiology, George Emil Palade University of Medicine, Pharmacy, Science, and Technology of Târgu Mureș, 38 Ghe. Marinescu Street, 540139 Târgu Mureș, Romania; 5Public Health Department, Carol Davila University of Medicine and Pharmacy, 050474 Bucharest, Romania; cls@g.unibuc.ro

**Keywords:** UVC disinfection, dental surfaces, material porosity, infection control, *Bacillus* spores, far-UVC, UV-LED

## Abstract

**Background/Objectives**: Environmental contamination of dental surfaces is a major vector for cross-infection. Ultraviolet-C (UVC) irradiation provides rapid, chemical-free decontamination; however, depending on wavelength and ventilation conditions, ozone generation may occur. This study evaluated the germicidal efficacy of UVC on three high-touch surfaces: a wooden work table, a stainless-steel consumables table, and a dental unit table. **Methods**: Surfaces were sampled at baseline, after 5 min (27 mJ/cm^2^), and after 10 min (54 mJ/cm^2^) of UVC exposure at 90 µW/cm^2^. Colony-forming units (CFU/cm^2^) were enumerated using Mueller–Hinton agar. **Results**: UVC achieved >99% reduction after 5 min and complete elimination after 10 min. Material properties (porosity, reflectivity, and grooves), along with quantified parameters like surface roughness (Ra) and contact angle, influenced minor differences in decontamination. **Conclusions**: Used with appropriate safety protocols, short-duration UVC irradiation effectively decontaminates dental surfaces and can complement chemical disinfection. Future studies must incorporate artificially soiled surfaces, biofilms, and emerging far-UVC/UV-LED technologies.

## 1. Introduction

Dental practices generate aerosols and splatter that readily deposit on surfaces, creating reservoirs for microorganisms, including *Staphylococcus aureus*, *Streptococcus mutans*, *Enterococcus faecalis*, *Pseudomonas aeruginosa*, and methicillin-resistant *S. aureus* (MRSA) [1,2,3,4]. Effective decontamination is essential to reduce cross-infection risk.

Chemical disinfectants (alcohols, quaternary ammonium compounds, sodium hypochlorite) remain routine but have drawbacks (surface compatibility, operator dependence, residues) [5]. Ultraviolet (UV) light—particularly UVC (100–280 nm; germicidal peak near 254 nm)—inactivates microbes by inducing pyrimidine dimers and other photoproducts in nucleic acids, preventing replication [6]. Modern approaches include low-pressure mercury lamps (254 nm), far-UVC (200–230 nm) sources, which may be safer for occupied spaces [7], UV-LEDs for compact systems [8], and pulsed xenon devices for short, high-dose exposures [9].

Although UVC has been validated in hospital settings for surface and air disinfection [6,7,8,9,10], comparative data on dental cabinet materials (wood, stainless steel, and dental unit plastic/composite) are limited.

Material characteristics, including porosity, surface roughness, UV reflectivity/absorbance, and structural features such as grooves or seams, critically influence UVC dose delivery, shadowing, and microbial sheltering [10,11]. Because these properties affect contaminant adhesion, microbial shielding, and secondary photon scattering, material-specific quantitative data are essential for developing practical dental decontamination protocols. Accordingly, beyond qualitative surface descriptions, this study incorporated quantitative measurements of roughness (Ra), wettability (contact angle), and spectral reflectance to assess material-dependent variability in UVC efficacy.

This study quantifies UVC efficacy on three representative dental cabinet surfaces, reports results as a dose–response (mJ/cm^2^), and evaluates how material properties may affect observed outcomes. We also contextualize our findings within emerging technologies (far-UVC, UV-LED) and address implications for safety and implementation in dental practice.

## 2. Materials and Methods

### 2.1. Study Design and Setting

An experimental study was conducted in a dental clinic treatment room. Three high-touch surfaces were selected: a varnished oak work table, a polished stainless-steel consumables table (AISI 304), and a dental unit table (medical-grade polymer composite with grooves).

### 2.2. Surface Characterization (Materials)

Because surface properties influence UVC effectiveness, each one was documented:-Wooden work table (oak, varnished)—Porosity and roughness: wood is inherently porous; varnishing reduces porosity, but micro-crevices and grain create micro-shadowing. UV interactions: organic components and pigments may absorb UVC, reducing penetration on micro-rough surfaces [11,12,13].-Stainless-steel consumables table (AISI 304, mirror/matte finish)—Reflectivity: polished stainless steel reflects a portion of incident UVC, increasing irradiance in some directions but also producing specular reflections and potential shadow cancelation; smooth surfaces minimize micro-sheltering [14,15,16].-Dental unit table (medical-grade plastic/composite with grooves and seams)—Topology and material composition: grooves and seams create shadowed regions; plastics typically have low UVC reflectance and may absorb UVC, while rougher finishes trap microorganisms in micro-crevices [17,18,19].

#### Quantitative Surface Property Assessment

To establish a direct correlation between material properties and UVC susceptibility, the following parameters were quantitatively measured on test squares adjacent to the microbial sampling areas before the UVC exposure trials:-Surface Roughness (Ra): The arithmetic average of the absolute values of the profile height deviations from the mean line (Ra) was measured in triplicate on each material using a calibrated profilometer (Mitutoyo Surftest SJ-210). Measurements to quantify micro-topography that can create “micro-shadowing” effects were taken across the grain of the wood, along the lay of the metal, and across a flat area on the dental unit polymer.-Contact Angle: The static contact angle was measured in triplicate for each surface using the sessile drop method with a goniometer (KRÜSS DSA100) and 5 μL of deionized water. This measurement indicates the surface’s hydrophobicity, which influences how contaminants and water-based aerosols adhere, potentially affecting the UVC dose received by microorganisms.-Spectral Reflectance (254 nm): The percentage of 254 nm UVC light reflected by each surface was measured using a calibrated spectroradiometer with an integrating sphere accessory. This helps determine how material reflectivity (e.g., for the stainless steel) might enhance the germicidal dose via scattered radiation.

### 2.3. UVC Device and Dosimetry

UVC irradiation was provided by a dual-lamp low-pressure mercury system (2 × 40 W, nominal emission at 254 nm) positioned 50 cm above the test surfaces. Irradiance was measured using a calibrated UVC radiometer, and uniformity was evaluated by recording measurements at three locations on each surface (center and two peripheral points). The measured irradiance at 50 cm was 90 µW/cm^2^. The UV source consisted of 254 nm lamps (Philips TUV 30W, Signify, Eindhoven, The Netherlands). Dose calculation:Dose (mJ/cm^2^) = Irradiance (µW/cm^2^) × time (s)/1000

-5 min (300 s): 90 × 300/1000 = 27 mJ/cm^2^-10 min (600 s): 90 × 600/1000 = 54 mJ/cm^2^

These intervals were selected to correspond to the rapid turnover times commonly required for surface disinfection between patient appointments in clinical dental practice.

Radiometer specifications and calibration—details of the instrument used to verify UVC lamp output are:-Instrument: Solar Light PMA2100 Radiometer (Serial Number: 2024-A-7215), Solar Light Company, Inc., Glenside, PA, USA. Spectral range: 200–280 nm-Measurement resolution: ±1 µW/cm^2^-Calibration: Traceable to the National Institute of Standards and Technology (NIST), and performed by them; most recent calibration performed on 25 March 2025. This calibration occurred one week before the experimental period.

UVC Lamp Specifications are:-Lamp type: Dual low-pressure mercury lamps (2 × 40 W)-Emission peak: 253.7 nm-Lamp height above samples: 50 cm-Irradiance at 50 cm: 90 µW/cm^2^ (mean of three measurements)-Operating hours at time of test: ~250 h

All experiments were performed under controlled ambient conditions (22.5 °C, 55% RH) The lamp was having a warm-up time of 3–5 min. Radiometer readings were rechecked periodically to confirm stable irradiance. This study did not include a comparison with emerging UVC-LED sources, which should be evaluated in future work for wavelength-specific efficacy differences. Figure 1 is a schematic photo of the experimental setup showing the lamp height (50 cm) used for bacterial reduction.

### 2.4. Sampling Protocol

Each surface was divided into nine 5 × 5 cm squares. Sampling was performed in triplicate from three adjacent squares for each condition and time point (baseline; after 5 min UVC; after 10 min cumulative UVC). These intervals correspond to typical dental operatory turnover times between patients. Triplicate sampling was chosen to ensure minimal variability and allow basic statistical comparison across conditions. Swabbing followed a standard approach aligned with ISO 18593 environmental surface sampling recommendations: sterile synthetic swabs, pre-moistened with sterile saline, stroked with firm rotational motion over the entire square area [20]. Swabs were placed into 1 mL sterile saline and vortexed for 10 s.

### 2.5. Culture and Identification

Aliquots (100 µL) of each sample were plated onto Mueller–Hinton agar using a sterile 10 µL loop and streaked in three directions to achieve semi-quantitative isolation. Plates were air-dried for 10 min and then incubated at 35 °C for 18 h. Colony-forming units (CFUs) were enumerated and converted to CFU/cm^2^ using the following calculation:CFU/cm^2^ = Colonies × 10/25.

The factor of 10 accounts for plating 100 µL from a 1 mL suspension, and the denominator reflects the 25 cm^2^ sampling area.

Selected colonies were Gram-stained to determine basic morphology. Observed isolates included Gram-positive cocci consistent with *Staphylococcus* spp. and a single Gram-positive rod, presumptively *Bacillus* spp. Full species-level identification (e.g., MALDI-TOF MS or 16S rRNA gene sequencing) was not performed in this pilot study and is recommended for future work to improve microbiological resolution.

Counts classified as “too numerous to count” (TNTC) were conservatively assigned a value of 220 CFU/cm^2^, corresponding to the maximum countable colony density for a 100 µL plated area, consistent with ISO microbiological enumeration guidelines.

### 2.6. Data Handling and Statistics

Counts > 200 CFU/cm^2^ (too numerous to count) were conservatively set to 220 CFU/cm^2^ for descriptive statistics. Mean ± SD and percentage reduction from baseline were calculated for each surface and condition. Differences across time points and materials were assessed using one-way ANOVA or, when normality assumptions were not met, the Kruskal–Wallis test, followed by Tukey’s post hoc comparisons. A significance level of α = 0.05 was applied. Statistical analyses were performed using SPSS v26.0 (IBM, Armonk, NY, USA).

## 3. Results

### 3.1. Baseline Contamination

To better contextualize the slight differences observed among wooden, metal, and composite dental-unit tables, we compiled literature values describing how surface microstructure and optical properties can affect UVC delivery to microorganisms (Table 1). These data help explain the modest residual contamination occasionally detected on polymer or grooved substrates.

All three surfaces exhibited high baseline contamination (>200 CFU/cm^2^), predominantly Gram-positive cocci consistent with *Staphylococcus* spp. No Gram-negative bacteria were cultured under these sampling conditions.

Quantitative measurements revealed significant material-dependent differences (Table 2).

### 3.2. Reduction After UVC Exposure

After 5 min (27 mJ/cm^2^), counts dropped markedly (>99%) on all surfaces; after 10 min (54 mJ/cm^2^), no colonies were observed except a single colony of a Gram-positive rod (presumptively *Bacillus*) on one dental unit sample. Figure 2 displays bacterial growth on Mueller–Hinton agar plates at three stages of UVC exposure. The images illustrate the reduction in colony-forming units (CFU) from baseline to post-exposure time points.

Figure 3 represents dose–response bar panels (three panels) showing mean CFU/cm^2^ (log scale) ± SD for each surface at Baseline, 5 min (27 mJ/cm^2^), 10 min (54 mJ/cm^2^).

Comparison across time points indicated a statistically significant reduction from baseline to 5 min and 10 min (*p* < 0.001). Exact *p*-values were below the software reporting threshold (<0.001), and thus presented as such. Differences between 5 and 10 min were small and not statistically significant for most surfaces due to near-zero counts (floor effect).

The data in Table 3 demonstrate the high germicidal efficacy of UVC light, achieving significant log reductions across all tested dental materials. A Log_10_ reduction of 2.0 corresponds to 99% bacterial elimination, while 3.0 corresponds to 99.9%. All surfaces achieved a Log_10_ reduction greater than 2.3 after just 5 min (27 mJ/cm^2^), indicating a bacterial load reduction of over 99.5%. The Wood and Metal surfaces showed Log_10_ reductions of ≥2.73 after 10 min, reflecting the elimination of nearly all vegetative bacteria down to the study’s limit of detection (LOD). The Dental Unit Polymer surface achieved the highest measured reduction at 2.85 Log_10_, effectively reducing the initial microbial population from over 213 CFU/cm^2^ to an average of just 0.3 CFU/cm^2^.

The limit of detection (LOD) of 0.4 CFU/cm^2^ derives from the plate conversion factor: One colony → CFU/cm^2^ = (10 mL × 1 colony)/25 cm^2^ = 0.4 CFU/cm^2^. Example of how LOD-based log reductions were calculated: For Wood, 10 min UVC (no colonies detected): log_10_(220.0/0.4) = 2.74, reported as ≥2.74.

## 4. Discussion

This study shows that UVC irradiation at 253.7 nm rapidly and substantially reduces culturable bacterial loads on three common dental cabinet surfaces. A dose of 27 mJ/cm^2^ (5 min at 90 µW/cm^2^) produced >99% reduction; 54 mJ/cm^2^ (10 min) eradicated vegetative bacteria in these experimental conditions, with the lone exception of a *Bacillus* colony attributed to spore resistance.

UVC kills microorganisms primarily by causing DNA/RNA photoproducts (pyrimidine dimers) that prevent replication and transcription [6]. Across all materials, UVC irradiation achieved rapid and substantial reductions in bacterial load, confirming its effectiveness as a rapid decontamination modality. Only after establishing this primary outcome do material characteristics help explain the minor differences in residual contamination observed among surfaces. Material properties can modulate UVC efficacy. Even varnished wood presents unique challenges for UVC disinfection due to its intrinsic microstructure. Microscopic grain and fissures can provide protective niches where microorganisms are shielded from direct irradiation, a phenomenon often called micro-shadowing. Organic wood constituents can absorb UVC radiation, reducing the effective surface dose [11,12,13]. Complete microbial elimination on wood was observed after 10 min of exposure. Nevertheless, subtle sheltering effects would likely become more pronounced under conditions involving biofilm formation or heavy soiling.

Stainless steel exhibits almost the opposite behavior, owing to its smooth, reflective properties. UVC light can undergo specular reflection, which enhances irradiance in angled or partially obstructed areas. However, this effect produces complex angular light distribution and can create secondary shadowing behind adjacent objects or at surface seams [14,15,16]. Thus, while reflectivity may support overall decontamination when lamp positioning is optimal, stainless steel remains susceptible to microbial persistence in shadowed microsites.

Plastics, such as those used in dental units, add further complexity. Surface grooves, seams, and non-uniform finishes form sheltered recesses where microorganisms persist despite UVC exposure. Moreover, plastics typically absorb a portion of UVC radiation, and the presence of organic plasticizers can alter light–surface interactions [17,18,19]. These combined factors suggest that effective decontamination of plastic components requires careful consideration of lamp angulation and, in many cases, the use of multiple lamp positions to minimize protected areas.

These considerations explain why proper lamp placement, avoidance of occluding objects, and possible multiple irradiance positions or longer exposure times are crucial for real-world effectiveness. The measured values support and clarify the microbial reduction results. The higher roughness of wood and polymer surfaces (Ra > 0.9 µm) is consistent with micro-shadowing that may shelter bacteria. Stainless steel, with minimal roughness (Ra ≈ 0.08 µm), provides fewer shelters, explaining its rapid decontamination. Contact angle differences reflected varying hydrophobicity, affecting microbial adhesion and droplet dynamics: stainless steel showed the highest hydrophobicity (83.6°), which may reduce microbial retention but could also influence how droplets spread or evaporate in clinical environments. Spectral reflectance at 254 nm was also highest for stainless steel (28.5%), indicating that secondary scattering could enhance local fluence, whereas the wood and polymer surfaces absorbed most incident UVC (<8%).

UVC provides rapid decontamination compared with chemical disinfectants, which require adequate contact time, full-surface wetting, and drying, and may cause corrosion or discoloration; staff compliance and environmental disposal also remain limiting factors [5,10]. UVC systems, when applied correctly, reduce dependence on consumable chemical products but must be implemented with safety protocols. However, sporicidal action poses an additional challenge as a difference to vegetative bacteria; spores typically require substantially higher UVC doses (often >100 mJ/cm^2^), combined modalities (e.g., heat + UVC), or chemical sporicides to achieve comparable reductions [11,21,22].

The detection of a single *Bacillus* colony after 10 min of UVC exposure reinforces this well-documented biological resistance: spores possess multiple protective layers, small acid-soluble spore proteins, and dipicolinic acid that stabilizes DNA and reduces photochemical damage. Bacterial spores can require 10–50 times the UVC dose needed for vegetative cell inactivation [23,24], owing to their protective architecture, including a dipicolinic acid (DPA)-enriched cortex and a modified DNA configuration that enhances resistance to photochemical damage [24].

Collectively, these findings show that quantitative surface parameters directly correlate with UVC susceptibility and should be integrated into decontamination protocol design.

The primary mechanism of UVC inactivation is the absorption of photons by DNA, leading to the formation of pyrimidine dimers that inhibit transcription and replication, ultimately leading to cell death [23]. However, some microorganisms can repair this damage through photoreactivation and dark repair, which can weaken the disinfection’s effectiveness [23,25]. Therefore, for an optimal germicidal action, the UVC dose must be sufficient to damage the DNA and overwhelm the organism’s repair mechanisms [23].

UVC disinfection presents a sustainable alternative or supplement to traditional chemical disinfectants. Unlike chemical agents, UVC leaves no residue and is chemical-free [25,26]. This reduces the environmental burden of chemical agents’ production, use, and disposal. The sustainability of UVC technology is further enhanced by the development of UV-LEDs, which offer a safer and efficient alternative to conventional mercury vapor lamps, eliminating hazardous materials [26]. This innovation also allows for operation at multiple wavelengths, providing greater versatility in disinfection [24,25].

The widespread adoption of UVC technology in healthcare necessitates strict adherence to safety standards. Direct exposure to UVC light harms human skin and eyes, potentially causing “sunburn” and photokeratitis [25,26]. Consequently, safety regulations mandate fail-safe systems to prevent lamps from operating when the device’s cover is removed or when personnel are present [8,25]. Recent innovations, however, include Far-UVC technology (around 222 nm), which is considered safe for use in occupied spaces as its short wavelength does not penetrate the skin’s outer layer or the tear layer of the eyes [24]. This allows for continuous decontamination without interrupting daily operations [24].

A comprehensive cost–benefit analysis is crucial for evaluating the adoption of UVC systems. While the initial investment in UVC devices can be significant, the technology offers a substantial return on investment by improving patient outcomes and operational efficiency [25,26,27]. UVC technology can reduce infection rates, shorten patient stays, and decrease readmission rates [26,27]. Moreover, it streamlines the disinfection process, reducing the time required for cleaning between procedures and minimizing human error [27]. One study found that UVC sanitizing devices were more cost-effective than relying solely on hand hygiene protocols, with a potential for significant long-term savings [28].

The findings of this study reinforce the value of a bundled approach to infection control, where UVC irradiation is used in conjunction with other methods. While UVC is highly effective on surfaces it can reach, its germicidal efficacy is limited by the “shadowing effect,” where areas not directly exposed to the light remain contaminated [26,29]. This makes it an ideal supplement to manual cleaning and chemical disinfection. Studies have shown that combining UVC light with antibacterial tablets can result in statistically superior disinfection outcomes [29]. This integrated strategy leverages the strengths of both methods, ensuring a more comprehensive and robust approach to infection prevention in dental practices.

Conventional 254 nm UVC can damage skin and eyes upon direct exposure (erythema, photokeratitis) [12,30]. Mitigation strategies include operation only in unoccupied rooms, safety interlocks, motion sensors, and shielded enclosures. Far-UVC (200–230 nm) has emerged as a promising alternative: studies show effective microbial inactivation with reduced penetration into the living human epidermis and cornea, suggesting lower risk for occupied spaces [7,22]. However, safety data are still under active investigation, and long-term human exposure limits remain under evaluation. UV-LED modules (varied wavelengths) offer compact form factors, longer life, instant on/off control, and no mercury content [8].

In addition to conventional mercury-lamp UVC systems, recent studies highlight alternative light-based disinfection technologies relevant to dental environments. UV-LED devices such as the optical-fiber prototype evaluated by Jeon et al. (2025) demonstrate effective bactericidal performance with advantages in portability and wavelength tunability, supporting their potential use on irregular or confined dental surfaces [17]. Pulsed xenon systems have also achieved significant reductions in healthcare-associated pathogens, delivering high-intensity, broad-spectrum UV bursts that shorten required exposure times [8]. Within dental settings, emerging investigations using UVC or far-UVC sources show comparable rapid inactivation of surface microorganisms, reinforcing that our findings align with the broader literature demonstrating the suitability of light-based technologies for chairside infection control.

The present study and the broader literature support UVC irradiation as an adjunct to conventional infection-control measures in dental environments. Our findings indicate that a short exposure to germicidal UVC can reduce surface contamination, provided dosing and operational factors are carefully controlled.

For routine, between-patient decontamination of flat work surfaces, an exposure of approximately 5 min at 50 cm (≥27 mJ/cm^2^) is sufficient to diminish the vegetative bacterial load. This duration aligns with reports demonstrating rapid inactivation of common clinical isolates at comparable fluence rates. When surfaces are likely to carry higher microbial burdens, such as after aerosol-generating procedures or in the presence of visible contamination, 10 min (≥50 mJ/cm^2^) or the combined use of chemical disinfectants and UVC is recommended to ensure deeper log reductions.

Operational considerations strongly influence efficacy. Surfaces should remain unobstructed during irradiation to avoid shadowing; where complex geometries are unavoidable, repositioning lamps or employing multiple lamp angles can help achieve uniform coverage. In rooms that remain occupied during decontamination, the use of far-UVC (207–222 nm) sources or shielded UV-LED enclosures may provide an additional safety margin, and conventional 254 nm devices should be equipped with interlocks or motion sensors to prevent accidental exposure.

Clinical surfaces are frequently coated with saliva, blood, pellicle, and well-established multispecies biofilms, all of which provide substantial protection against UVC penetration. Standardized soil loads—such as those described in ASTM E2197 [31]—or artificial saliva and blood matrices should be incorporated in future work to simulate clinically relevant organic burden. The absence of these factors in our experimental design creates an unavoidable clinical–experimental gap, and the present findings should therefore be interpreted as an upper-limit estimate of UVC efficacy under idealized conditions.

Another key microbiological limitation of this study is that organism identification was restricted to Gram staining, which offers only broad morphological information and cannot resolve bacteria at the species or even genus level. Species-level identification—ideally performed using MALDI-TOF mass spectrometry or 16S rRNA gene sequencing—would have enabled more precise characterization of surviving microorganisms, particularly the presumptive *Bacillus* colony detected following UVC exposure. Because bacterial identification relied solely on Gram stain morphology, organism-level resolution was not achieved. This limitation prevents the determination of whether clinically relevant oral pathogens were present, underscoring the need for MALDI-TOF MS or 16S rRNA sequencing in future studies, even for baseline microbial profiling.

Additionally, the exclusive use of aerobic culture conditions likely resulted in substantial underestimation of the true microbial diversity and baseline bioburden on the tested surfaces. Many clinically important oral bacteria—including obligate anaerobes associated with periodontal and endodontic infections—cannot be recovered under aerobic conditions and therefore would not have been detected.

Finally, no targeted assays were performed for highly relevant, disinfection-resistant dental pathogens such as *Enterococcus faecalis*, a species commonly implicated in persistent endodontic infections and known for its pronounced resistance to conventional antimicrobial protocols. Including such organism-specific testing would strengthen future evaluations of UVC efficacy in dental or clinical contexts.

Finally, sustainable implementation requires appropriate organizational measures. Staff should receive training on lamp positioning and fluence monitoring. Written protocols and checklists can support consistent practice, and maintenance records—particularly lamp operating hours and periodic radiometer checks—are essential to verify that devices continue to deliver the intended dose.

By incorporating these practical steps, dental teams can enhance environmental hygiene, reduce cross-contamination risks, and ensure safe and efficient use of UVC technology.

The current study has several limitations that should be considered before translating these findings into clinical practice. First, fungal and viral pathogens were not evaluated; therefore, conclusions are limited to bacterial decontamination. The experimental design—characterized by a small sample size (triplicates), absence of biofilm or heavy-soil models, and limited microbial identification and surface characterization—provides a preliminary framework but does not fully capture the complexity of a dental operatory environment. Although a 5 min UVC exposure shows potential as a safe adjunct surface decontamination measure between patient appointments, further clinical studies are needed to validate its effectiveness under routine practice conditions.

Methodological constraints should be acknowledged. Only culturable organisms were quantified, as molecular detection methods were not employed, and the use of triplicate samples limited statistical power. No deliberate organic soiling or biofilm models were incorporated, and species-level microbial identification was not performed. Reliance on naturally contaminated, otherwise clean surfaces—absent saliva, blood, or established biofilms—likely overestimates UVC efficacy relative to clinical settings, where complex organic deposits and structured biofilm communities provide substantial protection against ultraviolet disinfection.

The high efficacy observed with a 10 min UVC exposure produced a pronounced floor effect, with microbial counts approaching zero across nearly all tested surfaces. This limited residual variation constrained statistical analyses for material-dependent differences. To detect subtler effects, future investigations should increase replication (n > 3) or employ culture-independent detection methods, such as quantitative polymerase chain reaction (qPCR), which can quantify viable but non-culturable microorganisms. A larger sample size would enhance statistical power, narrow confidence intervals, and reduce variability inherent to environmental microbial sampling.

Another major limitation is the use of clean surfaces, which do not replicate biofilm formation or heavy organic soiling common in dental operatories [32,33]. Biofilms are complex, structured microbial communities encased in extracellular matrices that confer significant resistance to disinfection [32]. Future studies should incorporate artificially soiled substrates and established biofilm models to more accurately assess UVC efficacy under clinically realistic conditions. Microbial identification in this study was restricted to Gram staining, which provides only broad morphological classification without species-level resolution [34]. Advanced techniques, such as Matrix-Assisted Laser Desorption/Ionization–Time of Flight (MALDI-TOF) mass spectrometry or 16S ribosomal RNA (rRNA) gene sequencing, should be employed to enhance rigor. These methods enable precise species characterization and the detection of highly resistant pathogens, including Methicillin-resistant *Staphylococcus aureus* (MRSA) and *Enterococcus faecalis*, which pose significant risks for healthcare-associated infections [34,35]. Although Ra, contact angle, and spectral reflectance were quantitatively measured and reported in Table 2, these measurements were obtained under controlled laboratory conditions and may not fully capture the surface heterogeneity observed in clinical dental practice. Future studies should expand these quantitative measurements to a wider variety of dental materials under clinically contaminated conditions. Contact angle reflects surface hydrophobicity, which can influence contaminant adhesion and the UVC dose received, while spectral reflectance indicates how much UVC light is reflected; higher reflectivity can increase germicidal efficacy by scattering radiation into occluded areas [36,37].

The study focused on conventional UVC systems. Future research should explore emerging technologies, particularly far-UVC and UV-LED systems [7,8,29,38]. Far-UVC, with a shorter wavelength (e.g., 222 nm), is considered safe. Important parameters to assess include Surface Roughness (Ra), which can generate micro-shadows that protect microorganisms from UVC light in occupied spaces, presenting a potential strategy for continuous disinfection [35]. UV-LEDs are a more sustainable alternative to mercury lamps with hazardous materials [34].

Finally, a comprehensive cost–benefit analysis is crucial for justifying the widespread implementation of UVC technology in clinical settings. This analysis should compare the initial capital investment and energy costs of UVC deployment against the ongoing consumable costs of chemical disinfectants, factoring in labor, storage, and waste disposal [36]. While initial capital investment for germicidal UVC devices typically ranges from approximately €600 to €3000, these costs must be balanced against the recurring operational expenses of chemical disinfection, such as frequent purchases of pre-saturated wipes or sprays (e.g., estimated at €80–€150 per month per operatory). Furthermore, the electrical energy consumption of low-pressure mercury UVC systems is generally low (e.g., 40–80 W/h), contributing to lower long-term operating costs compared to the dependence on high-turnover chemical consumables. Such data will provide a robust economic argument for a more efficient, environmentally friendly disinfection strategy.

## 5. Conclusions

UVC irradiation rapidly inactivates vegetative bacteria on common dental surfaces, though bacterial spores remain resistant. Efficacy is strongly affected by surface porosity, roughness, and reflectivity, guiding lamp placement, and exposure time. These findings cannot be extrapolated to viruses or fungi without further study. Future work should assess biofilms, organic debris, and emerging Far-UVC or UV-LED systems. A five-minute UVC treatment can be safely used between patients as a practical adjunct to chemical disinfection.

## Figures and Tables

**Figure 1 dentistry-13-00596-f001:**
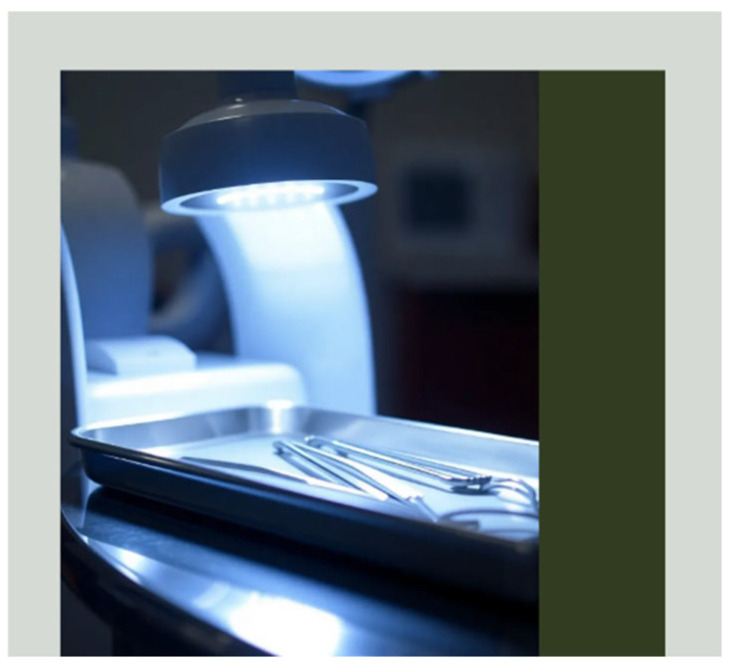
Experimental setup showing the UVC device positioned 50 cm above the test surfaces, lamp orientation.

**Figure 2 dentistry-13-00596-f002:**
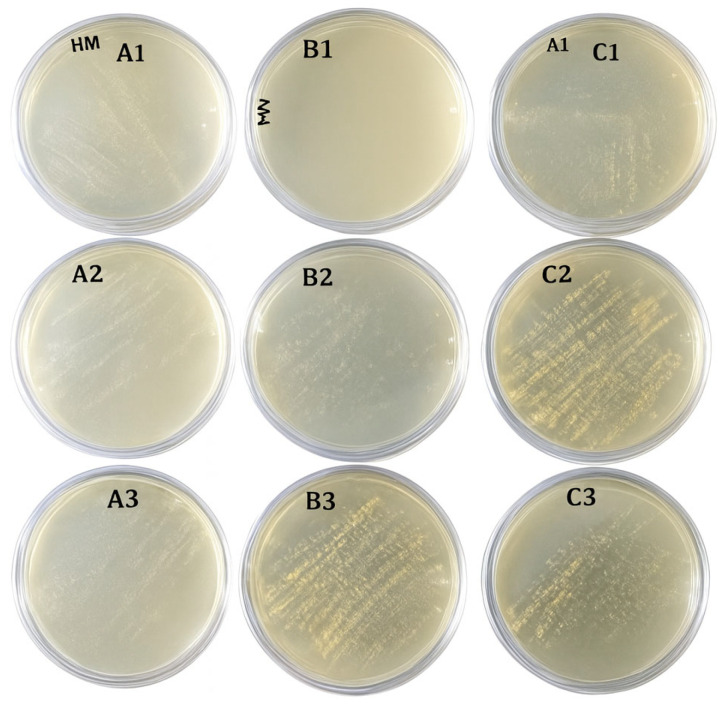
Bacterial growth on culture media at different stages of UVC exposure: (**A**) initial growth (baseline), (**B**) after 5 min of UVC exposure (27 mJ/cm^2^), and (**C**) after 10 min of UVC exposure (54 mJ/cm^2^). Each row represents triplicate assessments performed under identical conditions. **A1**, **A2**, **A3**, **B1**, **B2**, **B3**, **C1**, **C2**, and **C3** represent the triplicate sampling (1, 2, and 3) for the three different experimental conditions (A, B, and C) for a specific surface material. The figure shows these triplicates grouped by row.

**Figure 3 dentistry-13-00596-f003:**
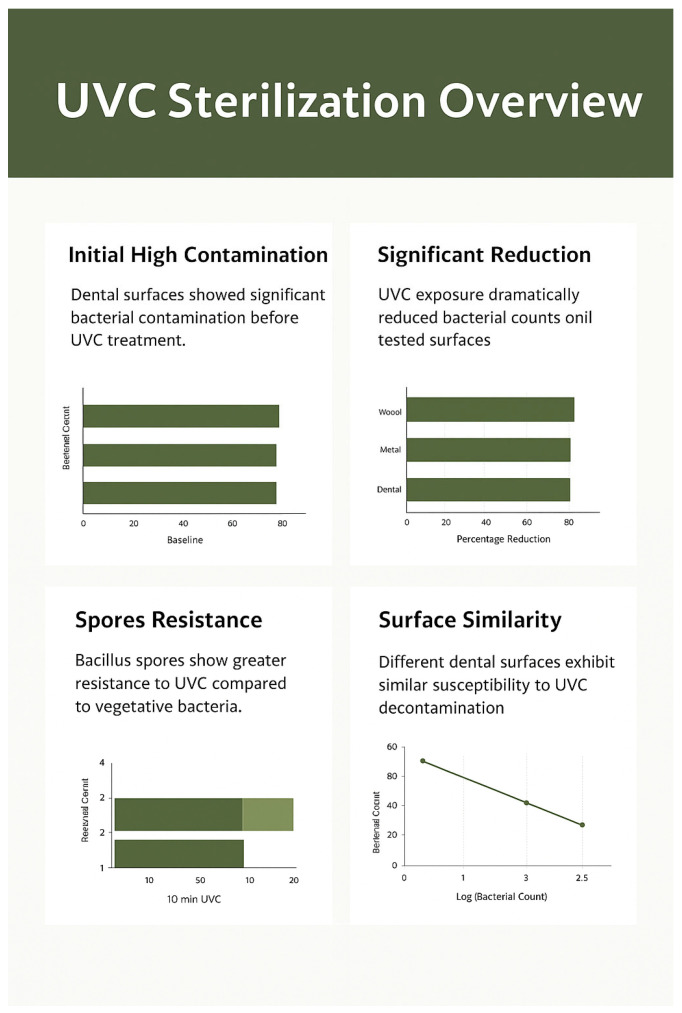
Bacterial load (CFU/cm^2^) on wooden, stainless-steel, and dental unit surfaces before and after UVC exposure (mean ± SD, n = 3). The y-axis log scale helps visualize baseline vs. post-treatment counts. The logarithmic y-axis emphasizes the drastic reduction from baseline (>200 CFU/cm^2^) to <2 CFU/cm^2^ after UVC exposure.

**Table 1 dentistry-13-00596-t001:** Estimated material characteristics influencing UVC efficacy (values from representative studies).

Surface	Porosity/Roughness	Typical UVC Reflectance (254 nm)	Likely Effect on Decontamination	References
Varnished oak (wood)	Moderate porosity; grain-aligned micro-crevices even after varnish	5–12% (varies with coating thickness and pigment)	Absorption and “micro-shadowing” may protect cells lodged in pores or along fibers	[11,12,13]
Stainless steel (AISI 304)	Ra ≈ 0.05–0.2 µm (polished)	25–30%	High reflectivity can enhance fluence on adjacent points; minimal micro-sheltering if free of debris	[14,15,16]
Dental-unit polymer/composite	Grooves and seams; matte areas	<8% (most plastics); varies by pigment	Absorption and irregular topology reduce direct fluence; seams may shield bacteria	[17,18,19]

Note: Values are averages from optical/engineering literature. Indicative values are provided here for literature comparison; measured values used in the analysis are presented in Table 2.

**Table 2 dentistry-13-00596-t002:** Quantitative Surface Properties of Tested Materials.

Material	Roughness (Ra)	Static Contact Angle	Spectral Reflectance (254 nm)
Wood (Varnished Oak)	1.5 ± 0.2 μm	68.4° ± 2.5°	7.2% ± 0.4%
Stainless Steel (AISI 304)	0.08 ± 0.01 μm	83.6° ± 1.8°	28.5% ± 1.2%
Dental Unit Polymer	0.9 ± 0.1 μm	76.2° ± 3.1°	5.1% ± 0.3%

Note: Values represent mean ± SD (n = 3). Ra measured using a contact profilometer; static contact angle measured using the sessile drop method; spectral reflectance measured at 254 nm using a UV–VIS spectrophotometer.

**Table 3 dentistry-13-00596-t003:** Mean Bacterial Load (CFU/cm^2^ ± SD), Log_10_ Reductions, and Statistical Significance for Each Surface Material Following UVC Exposure (n = 3).

Surface	Condition	Mean CFU/cm^2^ ± SD	Log_10_ Reduction	Significance vs. Baseline
Wood	Baseline	220.0 ± 10.0	—	—
Wood	5 min UVC	0.7 ± 0.6	2.5	<0.001
Wood	10 min UVC	0.0 ± 0.0	≥2.74	<0.001
Metal	Baseline	216.7 ± 7.6	—	—
Metal	5 min UVC	1.0 ± 1.0	2.34	<0.001
Metal	10 min UVC	0.0 ± 0.0	≥2.73	<0.001
Dental	Baseline	213.3 ± 7.6	—	—
Dental	5 min UVC	1.0 ± 1.0	2.33	<0.001
Dental	10 min UVC	0.3 ± 0.6	2.85	<0.001

Note: LOD (limit of detection) = 0.4 CFU/cm^2^; values below the LOD (reported as 0.0) have log_10_ reductions expressed as “≥” the value calculated using the LOD. Log_10_ reduction = log_10_(Baseline mean/post-treatment mean). Baseline values were not tested against themselves for significance.

## Data Availability

The raw data supporting the conclusions of this article will be made available by the authors on request.

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
