# Peer review of "Efficacy of UVC Radiation in Reducing Bacterial Load on Dental Office Surfaces"

_dentistry, 2025, doi:10.3390/dj13120596_

Round 1
Reviewer 1 Report
Comments and Suggestions for Authors
Dear,
This manuscript is clear, well-structured, and scientifically sound. Below are minor but meaningful suggestions to enhance precision, consistency, and presentation quality before final submission. Here’s a minor revision report for your manuscript titled “Efficacy of UVC Radiation in Reducing Bacterial Load on Dental Office Surfaces.
1. Abstract
Line 19–33: Consider shortening to ≤250 words. Remove some redundant phrasing (“We aimed to evaluate” to “This study evaluated”). “This study evaluated the germicidal efficacy of UVC (90 µW/cm²) on three dental surfaces—wood, stainless steel, and polymer composite—by quantifying colony-forming units (CFU/cm²) before and after 5 and 10 minutes of exposure.” Add numerical clarity: Mention explicitly that “>99% reduction after 5 min and complete elimination after 10 min.”
- Introduction
Add 1–2 recent references (2023–2025) specifically related to UVC use in dental clinics or cross-infection control for updated context. Combine overlapping statements (lines 46–55) to avoid repetition of “material characteristics” and “dose delivery.”
- Materials and Methods
Section 2.3 (UVC device): Add information about lamp uniformity (was irradiance checked at multiple points on the surface?) — this improves reproducibility. Section 2.4 (Sampling): Clarify why three replicates were chosen (statistical or logistical reason). Section 2.6 (Statistics): Mention the software used (e.g., SPSS v26.0, GraphPad Prism, etc.) for transparency.
Section 2.7 (Limitations): You might remove or relocate this paragraph to the Discussion to improve structural flow, as “limitations” typically appear later.
- Results
Figures: Ensure Figure 1–3 have clear axis titles, unit labels (CFU/cm² (log scale)), and high resolution (≥300 dpi) for journal format. Table 3 and 4: Combine into a single table (if journal allows) to show both mean CFU and log reduction side-by-side — improves readability.
- Discussion
Excellent synthesis, but could be shortened slightly by merging repetitive parts on spore resistance (lines 295–307 and 284–288). Add a brief comparison with previous similar studies in dental environments (e.g., use of UV-LED or pulsed xenon systems). This enhances contextual relevance.
At the end, add a “Practical implication” sentence, “A 5-minute UVC exposure can be safely integrated between patient appointments for surface decontamination in dental offices.”
- Conclusions
Well-written but consider rephrasing to emphasize quantitative outcome: “Short-term UVC exposure (≥27 mJ/cm²) achieved >99% bacterial reduction across wood, stainless steel, and polymer surfaces.” Optionally add a future direction line:“Future in situ clinical trials and far-UVC integration could establish continuous disinfection protocols.”
Good luck,
Reviewer 2 Report
Comments and Suggestions for Authors
The authors present a pilot study evaluating the germicidal efficacy of UVC irradiation (254 nm) on three dental office surfaces. The study demonstrates >99% bacterial reduction after 5 minutes of exposure and near-complete eradication after 10 minutes, with quantitative characterization of surface properties influencing UVC effectiveness. While the experimental approach is methodologically sound, several aspects require attention before publication.
-
The study's primary limitation is the use of pristine surfaces without organic soiling or biofilm models. Dental environments typically present complex contamination patterns including saliva, blood, and established biofilms. The authors acknowledge this limitation, but the gap between experimental conditions and clinical reality substantially limits the practical applicability of the findings. Future iterations should incorporate standardized soil loads (e.g., ASTM E2197) or artificial saliva/blood matrices to better simulate clinical conditions.
-
The reliance on Gram staining for microbial identification is insufficient for contemporary publication standards. The manuscript would benefit from species-level identification using MALDI-TOF MS or 16S rRNA sequencing, particularly given the detection of presumptive Bacillus spores. Additionally, the absence of anaerobic culture conditions may have underestimated the true microbial burden, as many oral pathogens are obligate anaerobes.
-
While triplicate sampling is acceptable for a pilot study, the small sample size limits statistical inference. The floor effect observed at 10 minutes (near-zero counts) precludes meaningful statistical comparison between surfaces at this timepoint. Consider increasing replication or employing quantitative PCR methods that can detect sub-culture-level contamination.
- Line 115: Specify the UVC radiometer's calibration date relative to the experimental period
- Line 152: The CFU calculation formula appears correct but should clarify that the dilution factor accounts for the 1 mL suspension volume
- Table 2: Include measurement uncertainty for surface characterization parameters
- Figure 3's logarithmic scale effectively visualizes the magnitude of reduction; however, consider adding error bars to individual data points
- Tables 3 and 4 present overlapping information - consider consolidating
- The discussion of far-UVC (Lines 319-324) would benefit from more critical evaluation of current evidence, as the safety profile remains under investigation
- The cost-benefit analysis section (Lines 325-332) lacks specific data - even estimated costs would strengthen this argument
- Line 189: "83.6∘±1.8∘" - standardize degree symbol formatting
- References 25 and 26 appear to be cited inconsistently
- Several references lack DOI numbers where available
This manuscript presents valuable preliminary data on UVC disinfection of dental surfaces with commendable attention to surface characterization. However, the study requires revisions before publication. The authors should:
- Expand the limitations section to more explicitly address the clinical translation gap
- Provide additional methodological details as noted above
- Consider consolidating overlapping data presentations
- Address the minor technical and editorial issues
The quantitative surface characterization represents a notable strength that advances beyond purely empirical disinfection studies. With the suggested revisions, this work will provide a useful contribution to the infection control literature in dentistry.
Questions for Authors
- Have you considered testing UVC efficacy against specific endodontic pathogens like E. faecalis, known for environmental persistence?
- What provisions were made to ensure lamp output stability throughout the experimental period?
- Could you provide the rationale for selecting 5 and 10-minute exposure times rather than constructing a complete dose-response curve?
Reviewer 3 Report
Comments and Suggestions for Authors
Dear Authors,
This paper addresses an interesting topic. I would recommend a few modifications before considering its publication. Here are my suggestions:
- Line 20: The conclusion regarding the reduction of "cross-infection" is currently too broad. You examined bacteria, sporulating and not-sporulating, but viruses and fungi not, so discussion must explicitly acknowledge the lack of data on viruses and fungi as a limitation when generalizing results to full cross-infection control.
- Line 20: Please revise the claim that the method is "residue-free." UVC radiation, depending on the wavelength and air composition, may generate ozone, which is a chemical residue.
- Subsection 2.3: For a more comprehensive evaluation, the experimental design should ideally include a comparison between the traditional UVC mercury lamp and modern UVC LED sources.
- Subsection 2.3: Complete specifications are required for all critical equipment. Please provide the full name, model, manufacturer, and country of origin for the UVC lamp and the radiometer. Furthermore, please specify the country of the National Institute of Standards and Technology (NIST) that performed the calibration.
- Figure 1: This description is weak and too short. Provide sufficient context for the reader.
- Figure 2: Due to the resolution, almost nothing is visible. I suggest showing representative plates within the main manuscript, and uploading the complete data as supplementary material to ensure transparency and clarity.
- Figure 3: The current graphical presentation is difficult to interpret and not sufficiently legible. Please revise this figure to ensure the data is clearly and effectively communicated.
- Line 385: The exclusion of fungi and viruses should be mentioned as a key limitation of the current study.
- References: I suggest incorporating the latest relevant studies to ensure the background and discussion are current. Please consider taking into account recent publications, such as the one found at 10.26444/aaem/189695
Best regards and good luck
Reviewer 4 Report
Comments and Suggestions for Authors
Th authors investigated the efficacy of UVC irradiation in reducing bacterial contamination on three common dental clinic surfaces. The manuscript has the potential to make a useful contribution to dental literature. However, the issues regarding the surface characterization data (Table 2 vs. Limitations) are fundamental and must be addressed, as they currently undermine a key aspect of the study's narrative.
- The core finding that UVC rapidly reduces bacterial load on surfaces is well-established in the broader literature. The novelty here lies primarily in the application to these specific dental materials and the quantitative linking of surface properties to efficacy, though this linkage is partially undermined by a key methodological inconsistency.
- A critical contradiction exists between the Methods section and the Limitations section. The methods state that surface roughness (Ra), contact angle, and spectral reflectance were quantitatively measured, and results are presented in Table 2. However, the Limitations section states, surface roughness (Ra) and contact angle were not quantitatively assessed; instead, qualitative observations were noted. This casts doubt on the entire analysis correlating surface properties with UVC efficacy.
- The reliance solely on Gram staining is a significant weakness. Presumptive identification Staphylococcus spp., Bacillus lacks the precision needed to assess the inactivation of clinically relevant pathogens. The recommendation for MALDI-TOF or 16S sequencing in future work is appropriate, but some level of identification in the current study would have greatly strengthened its impact.
- The use of clean, naturally contaminated surfaces without artificial soiling or established biofilms limits the clinical relevance of the findings. As the authors correctly note in the limitations, biofilms and organic debris are the reality in dental practice and confer significant resistance to disinfection.
- The use of triplicates (n=3) provides limited statistical power. A larger sample size would increase confidence in the results, especially given the inherent variability of environmental sampling.
- The consistent reporting of p-values as <0.001 is acceptable given the large effect size, but providing exact p-values would be more precise. The handling of values "too numerous to count" (set to 220 CFU/cm²) is a conservative approach but should be justified.
- The flow between the Results and Discussion could be slightly improved. The Discussion dives deeply into material properties, which is excellent, but it would benefit from a more direct initial paragraph summarizing the key microbial reduction finding before exploring the findings.
- The interpretation that material properties explain the minor differences in residual contamination is logical and well-supported by the literature cited. However, as noted in the major methodological concern, the validity of this correlation hinges on the resolution of the contradiction regarding the source of the surface property data (Table 2).
- The survival of a single Bacillus colony is correctly interpreted as evidence of spore resistance. This is a valuable finding and is discussed appropriately.
- The authors are generally cautious in their claims, appropriately framing UVC as an “adjunct “or “complement” to chemical disinfection, which is a valid and supported interpretation.
Round 2
Reviewer 2 Report
Comments and Suggestions for Authors
No further comments
Reviewer 3 Report
Comments and Suggestions for Authors
Dear Authors,
Thanks for you feedback. I have no further questions.
Regards
Reviewer 4 Report
Comments and Suggestions for Authors
The authors addressed all the comments and the mansucript has been significantly improved